# POST-HOC COMPRESSION OF LORA ADAPTERS VIA SINGULAR VALUE DECOMPOSITION

## ABSTRACT

Low-Rank Adaptation (LoRA) has become a widely used technique for personalizing text-to-image models such as Stable Diffusion. Although LoRA greatly reduces fine-tuning costs by introducing low-rank updates, practical deployments often involve one large backbone model combined with thousands of LoRA adapters. This creates a new challenge: even though each adapter is relatively small, the collective storage and transmission cost of large LoRA libraries becomes substantial. In this paper, we investigate whether existing LoRA adapters can be further compressed without retraining. We begin from the observation that while LoRA constrains updates to rank $r$ matrices, the effective rank of the learned updates is often significantly smaller, in practice, require only rank 1 to capture its expressive power. Motivated by this redundancy, we introduce a simple singular value decomposition (SVD)–based method to compress pretrained LoRA adapters. Through extensive experiments on a variety of text-to-image LoRA models, we show that trained LoRA models indeed contain considerable rank redundancy and SVD compression can consistently reduce adapter dimension without notable loss in generation quality.

## 1 INTRODUCTION

Recent text-to-image (T2I) diffusion models (Rombach et al., 2022; Podell et al., 2023; OpenAI, 2023; MidJourney) have achieved unprecedented success in generating diverse and high-quality visual content. These models have enabled a wide range of creative applications, from artistic rendering to product prototyping and character design. Yet, despite being trained on massive datasets, pretrained diffusion models remain confined to their training distribution and often struggle to synthesize novel concepts or faithfully replicate specific user-defined identities. For example, Stable Diffusion (SD) (Rombach et al., 2022) cannot natively generate a specific user's pet or a newly designed anime character. This limitation has driven the rapid development of personalization techniques, which adapt pretrained models to new concepts from only a handful of example images.

Existing personalization methods can be broadly grouped into encoder-based approaches and fine-tuning approaches. Encoder-based methods (Jia et al., 2023; Wei et al., 2023) aim to generalize across concepts without test-time adaptation, but typically require vast collections of text–image pairs and heavy computational resources, e.g., 1.4M pairs in InstantBooth (Shi et al., 2023) or 128 TPUv4 chips in SuTI (Chen et al., 2024). More practical alternatives directly fine-tune the base diffusion model using a small number of target images, with strategies such as text embedding learning (Gal et al., 2022), modifying cross-attention layers (Kumari et al., 2023), full-model fine-tuning (Ruiz et al., 2023), low-rank adaptation (LoRA) (Hu et al., 2021; cloneofsimo, 2022), or adjusting singular values of model parameters (Han et al., 2023). Among these, LoRA has become the de facto standard due to its efficiency, simplicity, and flexibility, enabling the community to rapidly create and share a massive ecosystem of LoRA adapters for diverse styles and concepts. Platforms such as Civitai now host tens of thousands of publicly released LoRA models, demonstrating the central role LoRA plays in real-world personalization pipelines.

Despite their popularity, LoRA adapters present two practical challenges. First, although LoRA introduces a low-rank decomposition into model weights, the effective rank of trained adapters may be far lower than their designed rank. For example, a LoRA adapter trained with rank $r = 8$ may, in practice, have an effective rank close to 1, suggesting substantial redundancy. Second, mod-

ern personalization practice increasingly involves maintaining one large backbone model alongside a growing collection of thousands of LoRA adapters. While each LoRA is relatively lightweight compared to full model fine-tuning, the cumulative storage and transmission burden of large LoRA libraries poses a serious challenge for scalable deployment, distribution, and mobile/edge applications.

Several recent works have attempted to reduce adapter size at training time, such as SVDiff (Han et al., 2023), PaRa (Chen et al., 2025), LyCoris (Yeh et al., 2024), and TriLoRA (Feng et al., 2024). These methods design improved parameterizations to encourage smaller or more stable adapters during fine-tuning. However, such approaches require training from scratch and therefore cannot be directly applied to the vast number of LoRA models that have already been released. This leaves open an important and practically relevant question: *can existing LoRA adapters be further compressed post hoc, without retraining?*

In this work, we answer this question affirmatively. We propose a simple yet effective singular value decomposition (SVD)–based compression method for pretrained LoRA adapters. Our approach decomposes the learned low-rank update into singular vectors, truncates to the top-$k$ components, and reconstructs a compressed adapter with reduced rank. This procedure is entirely training-free, maintains compatibility with existing LoRA implementations, and can be seamlessly integrated into current personalization workflows.

Our contributions are threefold:

- **Post-hoc compression strategy:** We introduce SVD-based rank reduction for LoRA adapters, requiring no retraining and preserving full compatibility with existing infrastructures.
- **Empirical verification of redundancy:** Through large-scale experiments on both the DreamBooth dataset and a diverse collection of Civitai LoRA adapters, we show that many adapters exhibit much lower effective rank than their nominal design, indicating significant redundancy.
- **Broad applicability:** We demonstrate that our compression strategy applies not only to vanilla LoRA but also to LoRA-style variants such as PaRa (Chen et al., 2025) and Ly-Coris (Yeh et al., 2024), paving the way for efficient storage, transmission, and deployment of large LoRA libraries.

Our findings highlight that the existing LoRA ecosystem can be made significantly more storage- and bandwidth-efficient, without sacrificing personalization quality. We envision this post-hoc compression strategy as an essential step toward scalable personalization at industrial scale, facilitating lightweight deployment, easier sharing, and sustainable growth of the LoRA community.

## 2 RELATED WORK

Diffusion-based generative models (Chang et al., 2023; Gu et al., 2022; Ho et al., 2020; Nichol & Dhariwal, 2021; Song et al., 2020a; Sohl-Dickstein et al., 2015; Song et al., 2020b) have achieved remarkable progress in image synthesis. The seminal DDPM (Ho et al., 2020) first demonstrated high-quality generation via iterative denoising, while score-based methods (Song et al., 2020a) improved efficiency by integrating score matching. These foundations have enabled large-scale text-to-image (T2I) systems such as DALL-E 2 (Ramesh et al., 2022), Imagen (Saharia et al., 2022), and Stable Diffusion (SD) (Rombach et al., 2022), which operate in latent spaces and leverage massive paired datasets for controllability and scalability. Among them, Stable Diffusion has become the most widely adopted open-source model and serves as the backbone for most personalization approaches.

Personalization methods can be broadly categorized into several directions. One line of work focuses on text embedding optimization, with Textual Inversion (Gal et al., 2022) learning new token embeddings to represent unseen concepts, though it often struggles with faithful alignment. Another stream finetunes the UNet directly, as in DreamBooth (Ruiz et al., 2023), with extensions such as Custom Diffusion (Kumari et al., 2023) and ConceptLab (Gandikota et al., 2023) seeking to balance alignment and generalization. Adapter-based approaches such as T2I-Adapter (Mou et al., 2024) and Adding Conditional Control (Zhang et al., 2023) introduce lightweight side modules to steer

generation. In contrast, training-free methods (Chen et al., 2024; Gal et al., 2023; Jia et al., 2023; Shi et al., 2023; Wei et al., 2023) avoid model updates altogether, typically at the cost of weaker personalization. More recent extensions investigate personalization in embedding space (Yeh et al., 2023) or across domains, for example DiffuseKrona (Marjit et al., 2025).

A particularly influential direction is parameter-efficient finetuning via matrix decomposition. LoRA (Hu et al., 2021; cloneofsimo, 2022; Gandikota et al., 2023) achieves this by injecting low-rank updates into pretrained weights. Instead of updating the full weight matrix, LoRA factorizes the update into two much smaller matrices whose product approximates the change. The pretrained weights remain frozen, and only these low-rank factors are trained, drastically reducing the number of trainable parameters while retaining sufficient expressiveness for personalization.

Several LoRA variants adopt alternative decomposition strategies. SVDiff (Han et al., 2023) applies singular value decomposition and fine-tunes the singular values of pretrained weights. PaRa (Chen et al., 2025) eliminates redundant rank components during training to improve stability and robustness. TriLoRA (Feng et al., 2024) integrates compact SVD into LoRA training, enhancing stability under long training horizons. LyCoris (Yeh et al., 2024) generalizes this idea further by combining multiple decomposition strategies, such as Kronecker products and additional low-rank structures, to increase flexibility while maintaining parameter efficiency. Despite their different formulations, these methods share the same goal: producing more compact adapters through rank reduction.

While these variants achieve smaller and often more stable adapters, their improvements in image quality compared to vanilla LoRA (Hu et al., 2021; cloneofsimo, 2022; Gandikota et al., 2023) are relatively modest. The primary benefit lies in reducing parameter counts and storage requirements rather than delivering substantial boosts in fidelity. Moreover, these approaches are all training-time modifications and therefore cannot be directly applied to the vast number of LoRA models that have already been trained and publicly released. This raises an important complementary question: *can existing LoRA adapters be further compressed post hoc?* Given the thousands of LoRA models that have already been deployed in practice, an effective compression method would directly reduce storage and transmission costs, providing significant practical value for scaling up personalization at industrial scale.

## 3 METHOD

### 3.1 REVISITING LORA

Low-Rank Adaptation (LoRA) (Hu et al., 2021) introduces low-rank updates into the linear transformation layers of a pretrained model. Let $W_0 \in \mathbb{R}^{d \times k}$ denote a frozen pretrained weight matrix. LoRA augments it with a trainable low-rank update $\Delta W \in \mathbb{R}^{d \times k}$ parameterized as

$$W = W_0 + \alpha \Delta W = W_0 + \alpha BA, \quad B \in \mathbb{R}^{d \times r}, \ A \in \mathbb{R}^{r \times k}, \tag{1}$$

where $r \ll \min(d, k)$ is the chosen rank and $\alpha$ is a scaling factor. This reduces trainable parameters from $O(dk)$ to $O(r(d + k))$, making LoRA particularly attractive for diffusion model personalization.

However, the designed rank $r$ does not necessarily reflect the *effective rank* of the learned update. In practice, even when $r = 8$ or $r = 16$, the singular value spectrum of $\Delta W = BA$ often decays rapidly, suggesting that most information can be captured by only a few dominant singular directions. This observation motivates further compression of trained LoRA adapters.

### 3.2 POST-HOC SVD COMPRESSION OF LORA

Given a trained LoRA adapter, we aim to reduce its effective rank without retraining. Although LoRA explicitly introduces a rank parameter $r$ during training, the actual update $\Delta W = BA$ often has a much lower effective rank in practice, as revealed by its singular value spectrum. This motivates us to apply singular value decomposition (SVD) as a tool for post-hoc analysis and compression.

**Step 1: Extracting the update.** From each trained LoRA adapter, we collect the update matrix $\Delta W = BA$. Note that this step is model-agnostic: regardless of which backbone or LoRA variant is used, the adapter can always be written as a low-rank update to a frozen weight matrix.

**Step 2: SVD factorization.** We perform singular value decomposition on $\Delta W$:

$$\Delta W = U\Sigma V^\top, \quad U \in \mathbb{R}^{d \times r}, \Sigma \in \mathbb{R}^{r \times r}, V \in \mathbb{R}^{k \times r}, \tag{2}$$

where the diagonal entries $\sigma_1 \geq \sigma_2 \geq \cdots \geq \sigma_r$ capture the energy distribution. Empirically, we observe that the singular values decay rapidly, implying that only a few directions dominate the adapter's effect.

**Step 3: Rank truncation and reconstruction.** To obtain a compressed adapter, we truncate $\Sigma$ to its top-$k$ singular values:

$$\Delta W_k = U_{:,1:k}\Sigma_{1:k,1:k}V_{:,1:k}^\top. \tag{3}$$

This gives the best rank-$k$ approximation in the Frobenius norm sense. We then reparameterize it back into the LoRA form:

$$\Delta W_k \approx B'A', \quad B' = U_{:,1:k}\Sigma_{1:k,1:k}^{1/2}, \quad A' = \Sigma_{1:k,1:k}^{1/2}V_{:,1:k}^\top, \tag{4}$$

so that the compressed adapter can be directly plugged into existing implementations without any modification to inference pipelines.

SVD provides the unique guarantee of producing the *best low-rank approximation* under both the spectral norm and Frobenius norm. This optimality ensures that information loss from compression is minimized, which is crucial for retaining the fine-grained personalization encoded in LoRA adapters. Moreover, SVD is numerically stable and well-supported in standard linear algebra libraries, making it straightforward to apply at scale.

The computational overhead of our compression step is minimal compared to LoRA training. Performing SVD on $\Delta W \in \mathbb{R}^{d \times k}$ costs $O(dk\min(d, k))$, but in practice $r \ll \min(d, k)$, so the cost is closer to $O(dkr)$. This is negligible relative to diffusion model finetuning, which requires thousands of gradient steps across billions of parameters. Thus, the compression can be executed efficiently even for large collections of LoRA models.

Our method can be summarized in three simple steps: (1) load the trained LoRA adapter weights, (2) perform SVD and truncate to the desired target rank $k$, (3) re-save the compressed adapter in the same LoRA format, as shown in Figure 1. This workflow requires no retraining, no modification to inference pipelines, and can be applied retroactively to any existing LoRA model in the community.

Although we adopt a uniform truncation rank $k$ across all layers, we observe that singular value spectra differ substantially across modules. We empirically analyze this in Section 4.5, which suggests opportunities for adaptive compression.

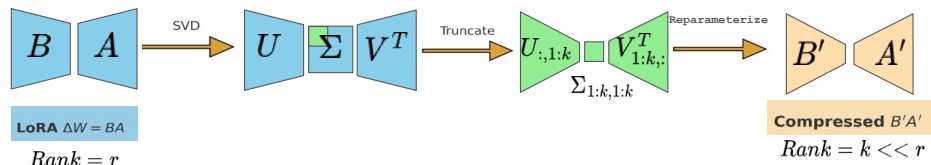

Figure 1: Overview of our post-hoc SVD compression. A trained LoRA adapter ($\Delta W = BA$) is decomposed via SVD, truncated to the top-$k$ singular values, and reconstructed as $B'A'$, resulting in a compressed LoRA adapter with rank $k \ll r$.

This post-hoc procedure highlights a unique strength compared to training-time approaches such as PaRa (Chen et al., 2025), LyCoris (Yeh et al., 2024), and TriLoRA (Feng et al., 2024). Those methods design new parameterizations to enforce compactness during finetuning, whereas our approach can be applied retroactively to any existing LoRA model. Given the rapidly growing ecosystem of community-trained LoRAs, this property is particularly valuable in practice, as it enables immediate compression of thousands of publicly released adapters without the need for retraining.

## 4 EXPERIMENTS

### 4.1 EXPERIMENTAL SETUP

We evaluate our proposed SVD-based compression method on several personalization settings. Our experiments consider both controlled benchmarks and real-world LoRA models that have been publicly released. Unless otherwise specified, Stable Diffusion v1.5 (Rombach et al., 2022) serves as the backbone. We compare vanilla LoRA (Hu et al., 2021), PaRa (Chen et al., 2025), and LyCoris (Yeh et al., 2024), as representative low-rank adaptation methods.

### 4.2 DATASETS AND EVALUATION

**Datasets.** For controlled training, we use the DreamBooth dataset (Ruiz et al., 2023), which consists of subject-specific images for personalization. In addition, we collect a large number of publicly available LoRA adapters from Civitai[1], covering diverse domains and styles.

**Evaluation metrics.** To assess personalization quality, we compute the CLIP image similarity scores $cos(\tilde{\mathbf{x}}, \mathbf{x})$ (Radford et al., 2021) between the training data and the generated image set (500 images).

For analyzing redundancy, we report singular value distributions across different layers. Finally, we report compression ratios and storage savings under various truncated ranks.

### 4.3 CLIP-BASED EVALUATION: TRAINING IMAGES VS. GENERATED IMAGES

We first train LoRA (Hu et al., 2021), PaRa (Chen et al., 2025), and LyCoris (Yeh et al., 2024) with the same rank dimension on the DreamBooth dataset. After training, we apply our SVD compression method to each adapter and compare: (1) the original trained model before compression, and (2) the compressed model with reduced rank. We report the CLIP cosine similarity averaged over multiple DreamBooth subjects. For each method, the reported standard deviation reflects the variation across different subjects rather than repeated runs.

As shown in Table 1, the drop in CLIP similarity after compression is very limited overall, indicating that SVD truncation effectively reduces adapter size while largely preserving personalization quality. We also provide qualitative examples of generated images before and after compression on subjects from the DreamBooth dataset in Figure 2.

A closer look at the table further reveals two interesting trends. First, when the designed rank $r$ is relatively large (e.g., $r = 16$), compressing directly to $r = 1$ leads to more noticeable decreases in CLIP similarity. Nevertheless, the values remain within an acceptable range, which aligns with our expectation that a higher initial rank introduces more redundancy but also amplifies the gap when reduced aggressively. Second, PaRa (Chen et al., 2025), which is designed to emphasize fidelity to training images, tends to achieve higher CLIP similarity before compression. However, this apparent advantage diminishes after compression, as the CLIP scores drop more sharply, highlighting that PaRa's strong memorization of training data may make it more sensitive to singular value truncation.

**Compression of LyCoris.** LyCoris (Yeh et al., 2024) extends LoRA by introducing a Kronecker product in the update formulation, where the forward pass is written as

$$h' = W_0 h + b + \gamma \Delta W h = W_0 h + b + \gamma [C \otimes (BA)]h, \tag{5}$$

with $C$ denoting an additional matrix introduced by LyCoris (Yeh et al., 2024). In our compression setting, we focus exclusively on the low-rank LoRA component $BA$. That is, we apply SVD to $\Delta W = BA$ and truncate its spectrum as described in Section 3, while leaving $C$ unchanged. This ensures that our method remains consistent with LyCoris' formulation, yet isolates compression to the same low-rank structure as in standard LoRA.

---

[1]https://civitai.com

Table 1: Average CLIP similarity (mean $\pm$ std) before and after SVD compression on the Dream-Booth dataset, evaluated at different LoRA ranks.

| Method | Rank | Original CLIP ↑ | Compressed CLIP ↑ | Drop |
|---|---|---|---|---|
| LoRA | 4 | $0.7424 \pm 0.0289$ | $0.7393 \pm 0.0321$ | -0.0031 |
| LoRA | 8 | $0.7504 \pm 0.0243$ | $0.7474 \pm 0.0261$ | -0.0030 |
| LoRA | 16 | $0.7912 \pm 0.0197$ | $0.7329 \pm 0.0303$ | -0.0583 |
| PaRa | 4 | $0.8203 \pm 0.0237$ | $0.7432 \pm 0.0241$ | -0.0771 |
| PaRa | 8 | $0.8339 \pm 0.0206$ | $0.7143 \pm 0.0251$ | -0.1296 |
| PaRa | 16 | $0.8624 \pm 0.0093$ | $0.6996 \pm 0.0246$ | -0.1628 |
| LyCoris | 4 | $0.7414 \pm 0.0391$ | $0.7294 \pm 0.0335$ | -0.0120 |
| LyCoris | 8 | $0.7435 \pm 0.0372$ | $0.7418 \pm 0.0385$ | -0.0017 |
| LyCoris | 16 | $0.7842 \pm 0.0294$ | $0.6921 \pm 0.0245$ | -0.0921 |

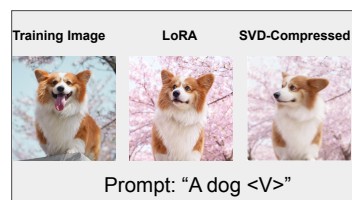 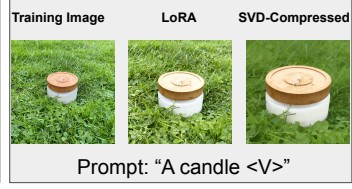 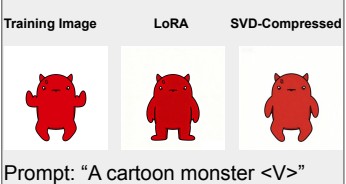

Figure 2: Qualitative comparison of LoRA and SVD-compressed LoRA generations on DreamBooth subjects. Each group shows (left) a training image, (middle) an image generated by the trained LoRA adapter, and (right) an image generated after SVD compression to rank $r = 1$. Prompts correspond to the subject's identifier tokens (e.g., "A dog <V>" or "A cartoon monster <V>"), consistent with the DreamBooth setup. We observe that compressed adapters produce results visually similar to those of the original LoRA, demonstrating that SVD compression preserves personalization quality while reducing model size.

## 4.4 CLIP-BASED EVALUATION: GENERATED IMAGES BEFORE AND AFTER COMPRESSION

To assess the impact of SVD compression, we compare CLIP cosine similarity between generated images from the original adapters and their compressed counterparts. We evaluate two scenarios: (1) different LoRA variants (LoRA, PaRa (Chen et al., 2025), LyCoris (Yeh et al., 2024)), and (2) different LoRA ranks (e.g., $r = 8, 4$).

For the DreamBooth dataset, we train adapters for each subject using LoRA, PaRa (Chen et al., 2025), and LyCoris (Yeh et al., 2024). After training, we compress each adapter to rank $r = 1$ using SVD, and for every subject we generate 20 images with both the trained adapter and its compressed version. CLIP cosine similarity is then computed between generated images and the training images, and results are averaged across all subjects.

For Civitai models, we directly use a large collection of publicly available LoRA adapters. Since training data is not available, we compare generations from the released models before and after compression, again generating 20 samples per model and reporting the average CLIP similarity across models.

The complete list of Civitai adapters used in this evaluation is provided in Appendix A. As shown in Table 2, the drop in CLIP similarity after compression is consistently very limited across both DreamBooth subjects and Civitai models, demonstrating that SVD truncation preserves semantic alignment while substantially reducing adapter size.

As shown in Table 2, the drop in CLIP similarity after compression is consistently very limited across both LoRA variants and LoRA models of different ranks, indicating that SVD truncation preserves semantic alignment while reducing adapter size.

Table 2: CLIP similarity (mean $\pm$ std) and model size before and after SVD compression. For the DreamBooth dataset, we train LoRA, PaRa (Chen et al., 2025), and LyCoris (Yeh et al., 2024) adapters on each subject and evaluate at compression from rank $r \in \{8, 4\}$ down to $r = 1$. For each subject, we generate 20 images using both the trained adapter and its compressed counterpart, then compute CLIP cosine similarity between generated and training images, and report the mean across subjects. For Civitai adapters, since no training images are available, we directly evaluate pretrained LoRA models and their compressed versions in the same way, reporting mean CLIP similarity across models.

| Dataset/Model | Setting | Backbone | CLIP ↑ | Model Size | Compressed Size |
|---|---|---|---|---|---|
| DreamBooth | LoRA ($r = 8$) | SDXL 1.0 | 0.8749 | 54.84MB | 7.29MB |
| DreamBooth | PaRa ($r = 8$) | SDXL 1.0 | 0.7868 | 19.52MB | 3.25MB |
| DreamBooth | LyCoris ($r = 8$) | SDXL 1.0 | 0.8483 | 129.52MB | 24.62MB |
| DreamBooth | LoRA ($r = 4$) | SDXL 1.0 | 0.8835 | 27.67MB | 7.29MB |
| DreamBooth | PaRa ($r = 4$) | SDXL 1.0 | 0.7938 | 10.68MB | 3.25MB |
| DreamBooth | LyCoris ($r = 4$) | SDXL 1.0 | 0.8537 | 72.48MB | 24.62MB |
| pastry | LoRA ($r = 32$) | SDXL 1.0 | 0.8284 | 217.87MB | 7.29MB |
| fried-egg | LoRA ($r = 32$) | SDXL 1.0 | 0.8036 | 217.87MB | 7.29MB |
| better-faces | LoRA ($r = 32$) | SDXL 1.0 | 0.8404 | 217.87MB | 7.29MB |
| huWoof | LoRA ($r = 8$) | SDXL 1.0 | 0.8730 | 54.76MB | 7.29MB |
| snorlax | LoRA ($r = 32$) | SDXL 1.0 | 0.8102 | 217.87MB | 7.29MB |
| tshirtdesignred | LoRA ($r = 16$) | SDXL 1.0 | 0.8943 | 162.64MB | 7.29MB |
| dpo-optim | LoRA ($r = 128$) | SDXL 1.0 | 0.8124 | 750.87MB | 7.29MB |
| cutecartoonred | LoRA ($r = 16$) | SDXL 1.0 | 0.7824 | 162.64MB | 7.29MB |
| emoji | LoRA ($r = 4$) | SDXL 1.0 | 0.7904 | 27.69MB | 7.29MB |
| aether-snow | LoRA ($r = 64$) | SDXL 1.0 | 0.8930 | 649.69MB | 7.29MB |
| oil-painting | LoRA ($r = 64$) | SDXL 1.0 | 0.7832 | 649.68MB | 7.29MB |
| lexus-lc | LoRA ($r = 32$) | SDXL 1.0 | 0.8274 | 217.87MB | 7.29MB |
| dinausaur-t-rex | LoRA ($r = 8$) | SDXL 1.0 | 0.8842 | 54.76MB | 7.29MB |
| arknights-chibi | LoRA ($r = 32$) | SDXL 1.0 | 0.8752 | 217.87MB | 7.29MB |
| 400gb | LoRA ($r = 64$) | SDXL 1.0 | 0.8275 | 485.95MB | 7.29MB |
| balloons | LoRA ($r = 32$) | SDXL 1.0 | 0.8942 | 217.87MB | 7.29MB |
| angel | LoRA ($r = 16$) | SDXL 1.0 | 0.8742 | 162.64MB | 7.29MB |
| eclipse | LoRA ($r = 32$) | SDXL 1.0 | 0.8962 | 217.88MB | 7.29MB |
| hogs | LoRA ($r = 128$) | SDXL 1.0 | 0.8842 | 860.02MB | 7.29MB |
| disney-princess | LoRA ($r = 32$) | SDXL 1.0 | 0.9174 | 217.93MB | 7.29MB |

## 4.5 SPECTRAL ANALYSIS OF LORA UPDATES

To better understand why post-hoc compression is feasible, we conduct a spectral analysis of the LoRA update matrices. For each LoRA module, we compute the singular values of its update $\Delta W = BA$ and sort them in descending order. Since the maximum rank of LoRA adapters in our collection is 32, we represent every spectrum as a fixed-length vector of dimension 32. If the effective rank $r < 32$, the missing entries are padded with zeros. This normalization enables direct averaging across different layers and models without additional binning.

We evaluate the singular value distributions using the collection of 20 publicly released LoRA models from Civitai, as shown in Figure 3. For each source, we aggregate results in several groups of layers: all layers (global average), time-embedding layers, cross-attention layers, the final high-level UNet blocks, convolutional layers, as well as the $K$ and $Q$ projection matrices in attention modules. Within each group, we compute the mean singular value at each index position $i \in \{1, \ldots, 32\}$, averaging across all corresponding layers and models.

Across all settings, we observe a rapid decay in singular values: the top singular values already account for the majority of the energy. These results provide direct evidence that LoRA updates contain substantial redundancy, and justify our approach of truncating the spectrum for post-hoc compression.

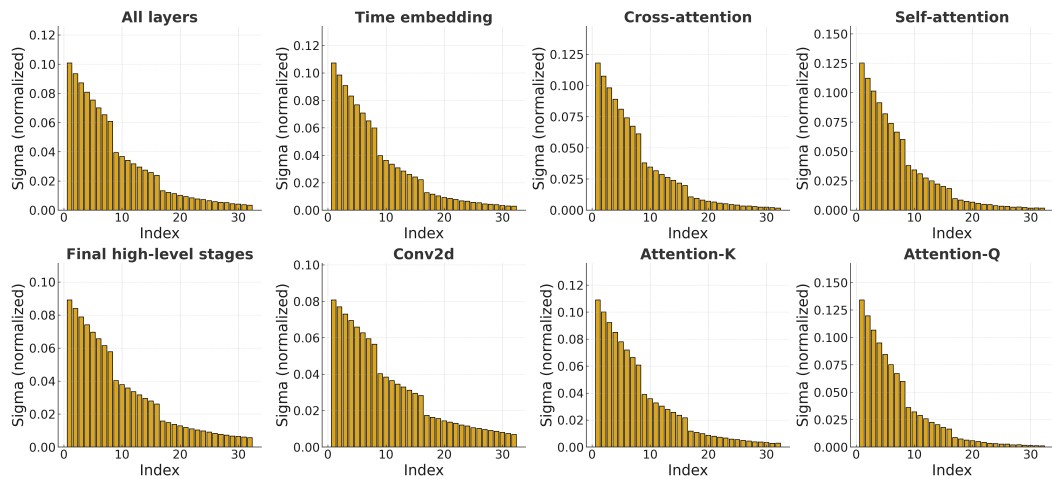

Figure 3: Average singular value spectra across different layer groups. Each subplot shows normalized singular values (descending order), averaged over models or subjects. Groups include: all layers, time embedding, cross-attention, self-attention, final high-level stages, convolutional layers, attention-$K$, and attention-$Q$. The pronounced spectral decay, especially in cross- and self-attention modules, indicates strong low-rank structure and high compressibility.

## 4.6 EXTENDED DEMONSTRATIONS OF COMPRESSED LORA ADAPTERS

Beyond quantitative benchmarks, we further verify whether compressed adapters still preserve the qualitative properties of standard LoRA models. Specifically, we examine two widely used features in community practice:

**Strength control via $\alpha$.** LoRA enables controlling the effect of an adapter during inference by scaling its contribution with a coefficient $\alpha$. We test this property on compressed adapters and observe that adjusting $\alpha$ smoothly interpolates between the base model generation and the target concept, indicating that compression does not harm the controllability of LoRA. As shown in Figure 4, varying $\alpha$ from 0 to 1 gradually shifts the generations from purely base-model outputs towards faithful reproductions of the personalized concept. This trend holds consistently across both Dream-Booth subjects (e.g., Candle) and diverse Civitai adapters (Pastry, Balloons, Emoji), demonstrating that controllability is well preserved even after SVD compression. (Figure 4).

**Multi-LoRA composition.** Another common application is fusing multiple LoRA adapters (e.g., one for a style and one for a character). We verify that compressed adapters can also be linearly combined during inference, yielding coherent generations that simultaneously reflect multiple concepts. For example, as shown in Figure 5, a style adapter (Emoji) and a content adapter (Balloons) can be mixed after compression, and the resulting generations preserve both properties without noticeable degradation. This demonstrates that post-hoc SVD compression preserves not only controllability but also the modularity of LoRA updates, enabling flexible composition across adapters.

For multiple adapters, standard LoRA composition takes the form

$$\Delta W_{\text{mix}} = \sum_{i=1}^{n} \alpha_i B_i A_i, \tag{6}$$

where $\alpha_i$ denotes the interpolation weight. After SVD compression, the same principle applies with the reparameterized factors:

$$\Delta W_{\text{mix},k} \approx \sum_{i=1}^{n} \alpha_i B_i' A_i'. \tag{7}$$

In all our experiments, as shown in Figure 5, we set $\alpha_i = 1$ for each adapter.

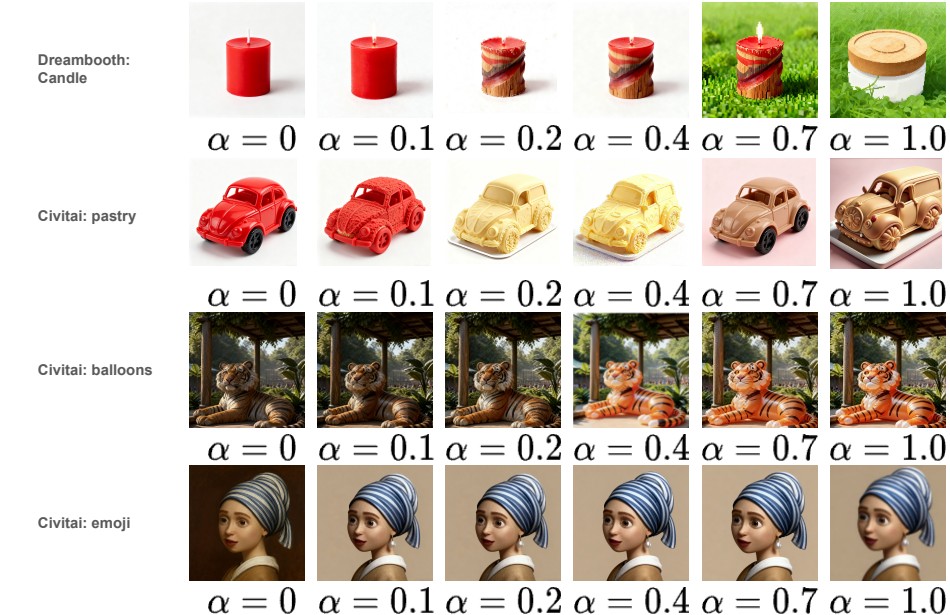

Figure 4: Controllability after compression. We vary the scaling factor $\alpha$ from $0$ (no adapter contribution) to $1.0$ (full adapter contribution) on compressed LoRA adapters. The results show smooth interpolation between base-model generations ($\alpha = 0$) and target-concept generations ($\alpha = 1$). Examples include a DreamBooth-trained adapter (Candle) and several community LoRA models from Civitai (Pastry, Balloons, Emoji). Across all cases, compressed adapters preserve the same degree of controllability as their uncompressed counterparts, confirming that post-hoc SVD compression does not compromise this key property of LoRA.

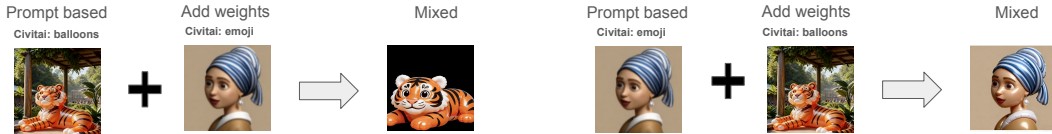

Figure 5: Multi-LoRA composition after compression. A style adapter (Emoji) and a content adapter (Balloons) are fused, and the generated images clearly retain both concepts, confirming that compressed adapters remain composable.

These results highlight that SVD-compressed adapters not only maintain quantitative alignment but also retain the practical usability of LoRA in real-world workflows, such as intensity control and multi-adapter fusion.

## 5 CONCLUSION

In this work, we investigated whether LoRA adapters, despite already being parameter-efficient, can be further compressed after training. We introduced a simple post-hoc method based on singular value decomposition (SVD), which decomposes the low-rank update matrices and truncates their effective rank. Our analysis revealed that trained LoRA adapters often exhibit substantial spectral redundancy, with only a few singular values carrying most of the information. These findings highlight that post-hoc compression is a practical and complementary direction to existing training-time variants such as PaRa, LyCoris. Beyond academic interest, the ability to shrink large libraries of LoRA adapters without retraining offers significant benefits for real-world deployment, reducing storage and transmission costs in industrial-scale personalization. Future work may explore adaptive rank selection across layers, and extending post-hoc compression to multimodal or cross-domain personalization tasks.

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

## ETHICS STATEMENT

This work adheres to the ICLR Code of Ethics. In this study, no human subjects or animal experimentation was involved. All datasets used were sourced in compliance with relevant usage guidelines, ensuring no violation of privacy. We have taken care to avoid any biases or discriminatory outcomes in our research process. No personally identifiable information was used, and no experiments were conducted that could raise privacy or security concerns. We are committed to maintaining transparency and integrity throughout the research process.

## REPRODUCIBILITY STATEMENT

We have made every effort to ensure that the results presented in this paper are reproducible. All code and datasets have been made publicly available to facilitate replication and verification. The experimental setup, including training steps, model configurations, and hardware details, is described in detail in the paper. We believe these measures will enable other researchers to reproduce our work and further advance the field.

## LLM USAGE STATEMENT

In preparing this paper, we made limited use of large language models (LLMs) such as ChatGPT as an assistive tool. The LLM was used for assisting with language polishing, grammar correction, and improving readability of draft text.

Importantly, all technical contributions, research ideas, experimental designs, and final analyses were conceived and verified by the authors. The LLM did not generate novel research concepts, perform data analysis, or contribute to the experimental results. The authors take full responsibility for all contents of the paper.

## A LIST OF CIVITAI LoRA ADAPTERS

In Figure 6, we provide qualitative results using 20 LoRA adapters obtained from the Civitai community. Table 3 summarizes the details of these adapters. Each model can be accessed at `https://civitai.com/models/<ID>` by replacing `<ID>` with the Civitai ID listed below.

## B ADDITIONAL RESULTS ON CIVITAI LoRA MODELS

To further demonstrate the practical utility of our post-hoc compression, we present qualitative examples from community-published LoRA adapters on Civitai. For each model, we compare the generations produced by the original adapter and its compressed counterpart. As shown in Fig. 6, the visual quality and semantic fidelity are well preserved after compression.

Table 3: Civitai LoRA adapters used in our experiments. Each adapter can be retrieved using its Civitai ID.

| Name | Civitai ID | Size |
|---|---|---|
| pastry | 189905 | 217.87 MB |
| fried-egg | 255828 | 217.87 MB |
| better-faces | 301988 | 217.87 MB |
| huWoof | 264013 | 54.76 MB |
| snorlax | 207790 | 217.87 MB |
| tshirtdesignredmond | 133031 | 162.64 MB |
| dpo-direct-preference-optimization | 242825 | 750.87 MB |
| cutecartoonredmond | 134720 | 162.64 MB |
| emoji | 144245 | 27.69 MB |
| aether-snow | 237531 | 649.69 MB |
| oil-painting | 118653 | 649.68 MB |
| lexus-lc | 375720 | 217.87 MB |
| dinausaur-t-rex | 1337587 | 54.76 MB |
| arknights-chibi | 984003 | 217.87 MB |
| 400gb | 143636 | 485.95 MB |
| balloons | 186256 | 217.87 MB |
| angel | 227519 | 162.64 MB |
| eclipse | 819155 | 217.88 MB |
| hogs | 603504 | 860.02 MB |
| disney-princess | 212532 | 217.93 MB |

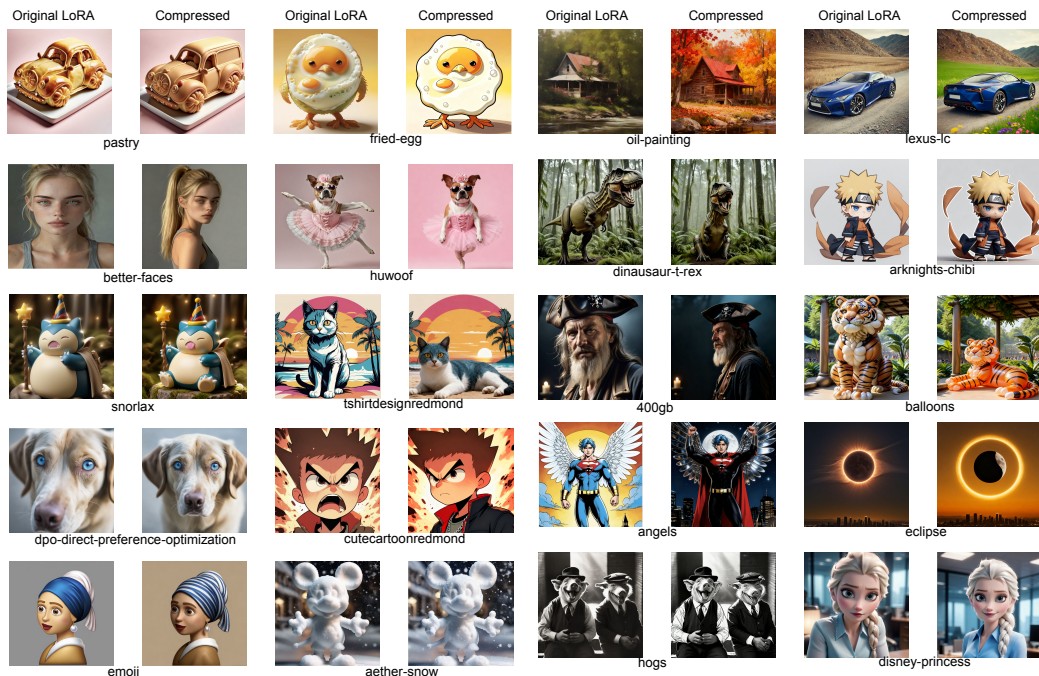

Figure 6: Generations from Civitai LoRA models before and after compression. Each pair shows results from the original adapter (left) and the compressed version (right). The compressed models retain visual quality while significantly reducing storage cost.

PROMPTS FOR EACH EXAMPLE

- **pastry** Positive: `car made out of pastry, food magazine shot, pastry <lora:pastry-sdxl:1>` Negative: `anime, cartoon, graphic,`

```
text, painting, crayon, graphite, abstract, glitch, blur,
bokeh
```

- **fried-egg** Positive: an adorable Sci-Fi Kiwi, digital art, made out of ral-friedegg `<lora:ral-friedegg:0.8>`, perfect composition, beautiful composition, dramatic, beautiful detailed, fantastic aesthetic, luxury, cute, aesthetic, warm light, lush Negative: N/A

- **better-faces** Positive: oil portrait of a pretty woman, 4ng3l face, beautiful girl, skinny, tank top, full lips, green eyes, black brows, freckles, blonde hair, straight hair, detailed skin, cinematic, great shading, moody lighting, neutral background, (by Lucian Freud:0.5), style of Jane Newland Negative: blurry, grainy, unfocused, low res, bad quality, nsfw, nude, naked, cartoon

- **huWoof** Positive: A HuWoof dressed as a ballerina is dancing `<lora:SDXL_HuWoof_LoRA:1>` Negative: blurry, grainy, unfocused, low res, bad quality, nsfw, nude, naked, cartoon

- **snorlax** Positive: Snorlax in hogwarts, wizard hat, wizard, magic wand. `<lora:snorlax_v3_1:0.8>` Negative: angry, leg up

- **tshirtdesignredmond** Positive: Illustrated T-shirt, Whimsical Cat Character, Sunset Beach Landscape, Tropical Color Palette, Fabric Texture, Bold Lineart, Dynamic Perspective, T shirt design,TshirtDesignAF, `<lora:CuteCartoonRedmond-CuteCartoon-CuteCartoonAF:1>` Negative: bad art, ugly, deformed, watermark, duplicated

- **dpo-direct-preference-optimization** Positive: RAW photo, a close-up picture of a dog, blue eyes, reflection in it's eyes Negative: bad art, ugly, deformed, watermark, duplicated

- **cutecartoonredmond** Positive: A angry boy, explosive scene, close-up, Cute Cartoon,CuteCartoonAF, `<lora:CuteCartoonRedmond-CuteCartoon-CuteCartoonAF:1>` Negative: bad art, ugly, deformed, watermark, duplicated

- **emoji** Positive: The girl with a pearl earring emoji `<lora:SDXL-Emoji-Lora-r4:0.2>` Negative: N/A

- **aether-snow** Positive: `<lora:Aether_Snow_v1_SDXL_LoRA:1.0>` full-body photo of mickey mouse made of snow, cinematic nightly Negative: N/A

- **oil-painting** Positive: mounting,flower,river,trees,wooden house,oil painting,art by Casey Baugh,art by sargent,out doors, bad hand,bad anatomy,floating hair,depth of field,anime,cartoon,over exposure,1 hand with many fingers,bad hand,extra hands and limbs,sack grid Negative: black and white,extra digit,static,thick eyebrows,thick lips,old women,missing hands and fingers,fusion limbs,multi girls,

- **lexus-lc** Positive: realistic photo of a blue lexuslc car on gravel road, outside, cinematic, `<lora:LexusLC_SDXL_v1:1>` Negative: N/A

- **dinausaur-t-rex** Positive: intricate photo of one single t-rex, prehistoric jungle, two claws at the hands of the short forelimbs, three claws at the each foot Negative: text, watermark, signatures, signs, emblems, words, numbers, ugly, blurry, anime, cartoon, painting, drawing, wrong proportions, disconnected limbs, malformed limbs, extra limbs, unfriendly, extra claws, extra legs, extra arms, wrong claws, wrong arms, wrong fingers, wrong legs

- **arknights-chibi** Positive: `<lora:ArknightsQ-000003:0.7>`,ArknightsQ, masterpiece,best quality,very aesthetic,absurdres,chibi,1 boy,Uzumaki Naruto,(full body),simple background,white background,solo,standing,closed mouth, Negative: EasyNegative, badhandv4, verybadimagenegative_v1.3, By bad artist -neg, ng_deepnegative_v1_75t,nsfw, lowres, (bad), text, error, fewer, extra, missing, worst quality, jpeg artifacts, low quality, watermark, unfinished, displeasing, oldest, early, chromatic aberration, signature, extra digits, artistic error, username, scan, abstract

- **400gb** Positive: cinematic film still (Digital Artwork:1.3) of (Ultrarealistic:1.3) `<lora:FF.102.colossusProjectXLSFW_49bExperimental.LORA:1.6>` photo of old pirate man, beard, fog, dark atmosphere, night, close up, cinematic shot, hard shadows, Fujifilm XT3, undefined,CGSociety,ArtStation . shallow depth of field, vignette, highly detailed, high budget, bokeh, cinemascope, moody, epic, gorgeous, film grain, grainy Negative: anime, cartoon, graphic, text, painting, crayon, graphite, abstract, glitch, deformed, mutated, ugly, disfigured, ((blurry)), worst quality, 3D, cgi, drawing, undefined,Scribbles,Low quality,Low rated,Mediocre,3D rendering,Screenshot,Software,UI,watermark,signature

- **balloons** Positive: balloonz, balloons, made out of balloons, tiger, standing tiger, balloon tiger, in the zoo, zoo made out of balloons `<lora:balloonz-sdxl:1>` Negative: N/A

- **angels** Positive: comic Angel, hero view, action pose, beautiful extremely detailed piercing eyes, scenery, detailed background, masterpiece, best quality, high quality, absurdres `<lora:angel_xl_v1:1>` . graphic illustration, comic art, graphic novel art, vibrant, highly detailed Negative: worst quality, low quality, poor quality, normal quality, medium quality, lowres, simple background, simple, ugly, photograph, deformed, glitch, noisy, realistic, stock photo

- **eclipse** Positive: `<lora:3cl1p53_01XL-000008:1.1>`,(3cl1p53 style),a RAW photograph of (3cl1p53 eclipse:0.9) (rising high in the sky above),with a (city skyline at night),surrounded by bright solar corona,HDR,(wide angle shot),sharp focus,(highly detailed),(8k wallpaper),intricately detailed,highres,absurdres,hyper realistic,8K UHD DSLR,IMAX,extremely intricate,4k textures,cinematic look),hyperdetailed Negative: N/A

- **hogs** Positive: film noir style hru, two pigs are smiling while sitting next to each other`<lora:hogs-cnt:1>` . monochrome, high contrast, dramatic shadows, 1940s style, mysterious, cinematic Negative: (worst quality, low quality, normal quality, lowres, low details, oversaturated, undersaturated, overexposed, underexposed, grayscale, bw, bad photo, bad photography, bad art:1.4), (watermark, signature, text font, username, error, logo, words, letters, digits, autograph, trademark, name:1.2), (blur, blurry, grainy), morbid, ugly, asymmetrical, mutated malformed, mutilated, poorly lit, bad shadow, draft, cropped, out of frame, cut off, censored, jpeg artifacts, out of focus, glitch, duplicate, (airbrushed, cartoon, anime, semi-realistic, cgi, render, blender, digital art, manga, amateur:1.3), (3D ,3D Game, 3D Game Scene, 3D Character:1.1), (bad hands,

```
bad anatomy, bad body, bad face, bad teeth, bad arms, bad
legs, deformities:1.3), ugly, deformed, noisy, blurry, low
contrast, realism, photorealistic, vibrant, colorful
```

- **disney-princess**    Positive:    `cinematic film still elsa, office lady<lora:add-detail-xl:1> <lora:princess_xl_v2:0.8>, . shallow depth of field, vignette, highly detailed, high budget, bokeh, cinemascope, moody, epic, gorgeous, film grain, grainy` Negative: `anime, cartoon, graphic, text, painting, crayon, graphite, abstract, glitch, deformed, mutated, ugly, disfigured`

