# OpenReview forum: "Post-hoc Compression of LoRA Adapters via Singular Value Decomposition"
_ICLR.cc/2026/Conference — Submitted to ICLR 2026_

### Official Review · Reviewer_AnW8 · 2025-10-18

**Soundness:** 2
**Presentation:** 4
**Contribution:** 4
**Rating:** 4
**Confidence:** 4

**Summary:**

This paper proposes a simple yet effective post-hoc compression technique for Low-Rank Adaptation (LoRA) adapters used in text-to-image diffusion models. While LoRA is already a parameter-efficient fine-tuning method, the authors observe that trained LoRA adapters often exhibit redundant rank components, i.e., their effective rank is much smaller than the nominal one. To exploit this redundancy, the authors apply Singular Value Decomposition (SVD) on the trained LoRA update matrices, truncate to the top-k singular values, and reconstruct a smaller adapter without retraining. Extensive experiments on DreamBooth and Civitai community LoRAs demonstrate that this post-hoc SVD compression achieves large reductions in storage (up to 90%+) while preserving CLIP-based similarity scores and visual quality.

**Strengths:**

- *Timely and practical contribution:* Managing large repositories of LoRA adapters is of imminent value to real-world deployment.
The proposed method is thus highly valuable for practitioners and large-scale personalization pipelines.
- *Simplicity and compatibility:* The SVD-based method is conceptually simple, theoretically grounded (optimal low-rank approximation), and fully compatible with existing LoRA implementations.
- *Comprehensive empirical validation:* The authors evaluate across multiple datasets (DreamBooth, Civitai), LoRA variants (LoRA, PaRa, LyCoris), and ranks. Both quantitative (CLIP similarity) and qualitative results are clear and convincing.
- *Practical impact and reproducibility:* Demonstrations such as α-scaling (controllability) and multi-LoRA composition confirm usability after compression. The authors commit to open-sourcing code and datasets for reproducibility.

**Weaknesses:**

- *Limited theoretical depth:* While the SVD rationale is sound, the paper remains largely empirical. There is no theoretical analysis of why LoRA training yields such rapid spectral decay or how compression affects gradient directions in the fine-tuned subspace.
- *Evaluation scope:* The evaluation relies heavily on CLIP similarity, which may not fully capture perceptual or artistic fidelity. More human or FID-based evaluations could strengthen claims about “no notable loss in generation quality.”
- *Fixed rank truncation:* The method uses a uniform rank across layers, even though the singular value spectra differ significantly.
An adaptive or energy-based rank selection criterion could yield better trade-offs between quality and compression.
- *Lack of runtime evaluation:* While storage reduction is well demonstrated, it would be useful to quantify potential improvements in inference latency, memory footprint, or mobile deployment feasibility.
- *Positioning vs. prior work:* The paper could more explicitly compare post-hoc SVD compression with training-time compactness methods (SVDiff, TriLoRA, PaRa) under matched compression ratios to highlight relative benefits or trade-offs.

Minor:
- A schematic illustration of how SVD compression integrates into LoRA loading code would be helpful.
- Line 169: "where the diagonal entries $\sigma_1$" - by this, I assume "diagonal entries of $\epsilon$"?

**Questions:**

1. How sensitive is the compressed LoRA quality to the chosen truncation rank k?
2. Could an adaptive criterion based on explained variance (e.g., retaining 95% of singular value energy) yield more consistent results?
Does SVD compression affect downstream finetuning or composition with other adapters (e.g., VeRA)?
3. Could the authors provide insights into per-layer rank redundancy—which layers or modules (e.g., cross-attention vs. convolution) are most compressible?
4. Have the authors considered a hybrid approach, where SVD-compressed adapters are fine-tuned briefly to recover any lost quality?

**On broader impacts of the proposed method:**

- What concerns me is the broader applicability of this test-time method to LoRA adapters for other downstream tasks. Can the authors provide results on standard classification task(s) if the rebuttal time period allows for this?
- *Thoughts on LoRA merging*: Given that the proposed test-time LoRA compression method preserves only few top-k singular values, effectively reducing the rank to 1, can the authors comment if this method could be applied for effective LoRA merging as well? It would be interesting to see how multiple rank 1 LoRA adapters (obtained by the proposed decomposition method) be composed/merged together to capture meaningful downstream task semantics by not accumulating errors - this could clearly impact the wider deployment of the proposed compression method.

---

### Official Review · Reviewer_suHJ · 2025-10-31

**Soundness:** 3
**Presentation:** 1
**Contribution:** 1
**Rating:** 2
**Confidence:** 4

**Summary:**

In my understanding, this method compresses LoRA adapters after training by using SVD. It takes the learned update matrix, decomposes it into singular values, keeps only the top few, and rebuilds a smaller adapter. This reduces rank and storage without retraining, while keeping most of the personalization quality. It is a straightforward approach.

Here are the authors claimed contributions:
Their post-hoc compression strategy.
Empirical evidence of redundancy.
Showing that this is broadly applicable.

**Strengths:**

Clear idea: The paper explains the SVD-based compression method in a straightforward way, making it easy to understand and implement.
Empirical backing: The paper shows experiments and spectral analysis to demonstrate that LoRA adapters have significant rank redundancy, supporting the feasibility of post-hoc compression.

**Weaknesses:**

Here are the major weaknesses of this approach:
1. The contribution is not novel.
2. The authors observations have already been demonstrated in previous work.
3. The experimental analysis does not contain any comparison with related work.
4. The authors don't discuss what is the motivation to reduce the rank of an adapter after training. In most solutions, the adapters are merged into the model during inference. If there's no inference saving or training compute saved, how is the method efficient?


There exists a lot of related work not discussed, which have already come to these conclusions and the approach which is presented as novel has already been implemented in libraries such as Kohya and Peft. Some of the works are listed below. The authors are encouraged to perform a thorough literature review.

Post-hoc Compression Methods:

PHLoRA: Post-hoc Low-Rank Adapter Extraction from Full-Rank Checkpoints.
Compress-then-Serve: Efficient Multi-LoRA Serving via Shared Low-Rank Basis.
KnOTS: Knowledge Transfer via Orthogonal Subspaces for LoRA Merging.
DyLoRA: Dynamic Low-Rank Adaptation for Parameter-Efficient Fine-Tuning.


Works Highlighting Redundancy in Low-Rank Dimensions:

SARA: Singular-Value Based Adaptive Low-Rank Adaption
FouRA: Fourier Low-Rank Adaptation
SVDiff: Compact Parameter Space for Diffusion Fine-Tuning
PaRa: Personalizing Text-to-Image Diffusion via Parameter Rank Reduction
TriLoRA: Integrating SVD for Advanced Style Personalization in Text-to-Image Generation
LyCoris: Navigating Text-to-Image Customization with Flexible Low-Rank Structures

**Questions:**

What is the motivation to reduce the rank of an adapter after training?

---

### Official Review · Reviewer_HaoF · 2025-11-02

**Soundness:** 2
**Presentation:** 2
**Contribution:** 1
**Rating:** 2
**Confidence:** 5

**Summary:**

This paper proposes a training-free method for compressing LoRA adapters. The core idea is to apply singular value decomposition (SVD) to the LoRA parameters and reconstruct the adapters using a small number of singular components. The method aims to maintain model performance while significantly reducing the number of trainable or stored parameters. Extensive experiments are conducted to demonstrate the efficiency–effectiveness trade-off.

**Strengths:**

The method is simple, intuitive, making it easy to implement in existing LoRA-based systems.
The presentation is clear and concise, allowing readers to easily understand the procedure and reproduce the approach.

**Weaknesses:**

1. The paper does not sufficiently justify the use of SVD as the decomposition technique. Why is SVD preferred over alternative low-rank approximation methods (e.g., randomized projections, or tensor decompositions)? Additionally, the choice of using rank-1 compression appears subjective and is not empirically validated. A broader exploration of decomposition and rank selection strategies would strengthen the technical contribution.

2. According to Figure 3, the spectral decay of the LoRA updates is not particularly sharp, suggesting that the weight updates may not be strongly low-rank. This observation undermines one of the central assumptions of the method. A more detailed analysis or justification of this assumption is needed.

3. The presentation of results in Table 1 is unclear. It is not obvious which method is the proposed one, and whether LoRA, PaRa, or LyCoris serve as baselines or components of the method. Clearer labeling and explanation are required to interpret the results properly.

**Questions:**

see Weaknesses

---

### Meta-Review · Area_Chair_LJ61 · 2026-01-06

**Summary:**

The concerns are on the novelty, experimental analysis, and theoretical depth of this paper.

**Reviewer Concerns:**

No rebutal is entered so they are still outstanding.

**Reviewer Scores:**

None.

---

### Decision · Program_Chairs · 2026-01-26

Reject